# Comparing statistical learning methods for complex trait prediction from gene expression

**Noah Klimkowski Arango** [1,2], **Fabio Morgante** [1,2]*

**1** Center for Human Genetics, Clemson University, Greenwood, SC, United States of America, **2** Department of Genetics and Biochemistry, Clemson University, Clemson, SC, United States of America

* fabiom@clemson.edu

## Abstract

Accurate prediction of complex traits is an important task in quantitative genetics. Genotypes have been used for trait prediction using a variety of methods such as mixed models, Bayesian methods, penalized regression methods, dimension reduction methods, and machine learning methods. Recent studies have shown that gene expression levels can produce higher prediction accuracy than genotypes. However, only a few prediction methods were tested in these studies. Thus, a comprehensive assessment of methods is needed to fully evaluate the potential of gene expression as a predictor of complex trait phenotypes. Here, we used data from the *Drosophila* Genetic Reference Panel (DGRP) to compare the ability of several existing statistical learning methods to predict starvation resistance and startle response from gene expression in the two sexes separately. The methods considered differ in assumptions about the distribution of gene effects—ranging from models that assume that every gene affects the trait to more sparse models—and their ability to capture gene-gene interactions. We also used functional annotation (*i.e.*, Gene Ontology (GO)) as a source of biological information to inform prediction models. The results show that differences in prediction accuracy exist. For example, methods performing variable selection achieved higher prediction accuracy for starvation resistance in females, while they generally had lower accuracy for startle response in both sexes. Incorporating GO annotations further improved prediction accuracy for a few GO terms of biological significance. Biological significance extended to the genes underlying highly predictive GO terms. Notably, the Insulin-like Receptor (*InR*) was prevalent across methods and sexes for starvation resistance. For startle response, crumbs (*crb*) and imaginal disc growth factor 2 (*Idgf2*) were found for females and males, respectively. Our results confirmed the potential of transcriptomic prediction and highlighted the importance of selecting appropriate methods and strategies in order to achieve accurate predictions.

## Introduction

Predicting yet-to-be observed phenotypes for complex traits is an important task for many branches of quantitative genetics. Complex trait prediction was developed in agricultural

**Data Availability Statement:** All DGRP lines are available from the Bloomington Drosophila Stock Center (Bloomington, IN). All raw and processed RNA-Seq data are available at the NCBI Gene

Expression Omnibus (GEO; https://www.ncbi.nlm.nih.gov/geo/) under accession number GSE117850. Phenotypic data are available at http://dgrp2.gnets.ncsu.edu/. The code used for the analyses is available at https://github.com/nklimko/dgrp-starve.

**Funding:** Research reported in this publication was supported in part by the National Institute of General Medical Sciences of the National Institutes of Health under Award Number R35GM146868 to FM. The content is solely the responsibility of the authors and does not necessarily represent the official views of the National Institutes of Health. The funders had no role in study design, data collection and analysis, decision to publish, or preparation of the manuscript.

**Competing interests:** The authors have declared that no competing interests exist.

breeding to select the best performing individuals for economically important traits such as milk yield in dairy cattle using estimated breeding values (EBVs). While EBVs have been traditionally computed using pedigree information, the availability of genotyping arrays has made it possible to compute genomic EBVs (GEBVs) [1, 2]. GEBVs are linear combinations of the genotypes of the target individuals and the effect sizes of many genetic variants along the genome computed in a reference population. The same concept has later been applied to human genetics, especially in the context of precision medicine. Here, the goal is to predict medically relevant phenotypes such as body mass index (BMI) or disease susceptibility using Polygenic Scores (PGSs) [3, 4]. While GEBVs and PGSs are technically the same, the different goals of these two quantities (*i.e.*, selection for GEBVs and prevention/monitoring for PGSs) have important implications. We refer the readers to [2] for a comprehensive treatment of this topic.

The estimation of the effect sizes of genetic variants to be used for prediction can be done using a variety of methods. The most common methods have regression at their core, where the response variable is the phenotype of interest and the predictor variables are the genotypes for a set of genetic variants [5]. Since the number of genetic variants, $p$, is usually much larger than the sample size, $n$, (the well-known $p >> n$ problem in statistics [6]), methods that perform variable selection or regularization of the effect sizes are needed. These methods encompass dimension reduction methods (*e.g.*, principal components regression), penalized regression methods (*e.g.*, ridge regression), linear mixed models (*e.g.*, GBLUP), Bayesian methods (*e.g.*, BayesC), and machine learning (*e.g.*, random forest) [7–11]. These methods differ in the assumptions they make regarding the distribution of the effect sizes, with some methods performing only effect shrinkage and some methods performing variable selection as well [5]. Research focused on comparing several methods has shown that there is no single best method, with performance being affected by the genetic architecture of the trait of interest (*e.g.*, sparse vs dense), the biology of the species (*e.g.*, the extent of linkage disequilibrium), and the assumptions of the method [12, 13].

Traditionally, genotype data has been used for complex trait prediction since they are easy and cost-effective to obtain. However, it is now possible to obtain multidimensional molecular data such as gene expression or metabolite levels at a reasonable cost. This advancement has made the use of additional layers of data for complex trait prediction feasible. Broadly, genetic information flows from DNA to RNA to proteins, and then to metabolites, which affect phenotypes [14]. Using these intermediate layers of data could improve prediction accuracy for some traits. In addition to being biologically 'closer' to phenotypes, these molecular data can be thought of as endophenotypes, which are affected by environmental conditions as well as genetic effects. Thus, endophenotypes could capture environmental and gene-by-environment effects [15].

Recent work has shown that using additional omic data types can result in more accurate predictions [15–20]. In particular, transcriptomic data has shown good potential for improving prediction accuracy. For example, Wheeler *et al.* used lymphoblastoid cell line data to show that gene expression levels provided much higher accuracy than genotypes when predicting intrinsic growth rate [16]. Morgante *et al.* found that prediction accuracy of starvation resistance in *Drosophila melanogaster* was higher when using gene expression levels instead of genotypes [18].

While these studies have shown the potential of gene expression as a predictor of complex phenotypes, only a few statistical methods were used, with most studies using linear mixed models. Studies using genotypes have found that prediction accuracy may vary substantially depending on the method used. However, a comprehensive comparison of methods for transcriptomic prediction is currently missing (to the best of our knowledge). Thus, in this study,

we sought to compare several common methods spanning dimension reduction, penalized linear regression, Bayesian linear regression, linear mixed model, and machine learning in their ability to predict starvation resistance and startle response from gene expression levels using data from the *Drosophila* Genetic Reference Panel [21].

## Materials and methods

### Data processing

The *Drosophila melanogaster* Genetic Reference Panel (DGRP) is a collection of over 200 inbred lines that have full genome sequences and phenotypic measurements for several traits [22]. Additionally, prior work from [23] obtained sex-specific whole-body transcriptome profiles for 200 DGRP lines by RNA sequencing. We followed the steps described in [18] to filter for genetically variable and highly expressed genes. This filtering process resulted in 11,338 genes in females and 13,575 genes in males.

In this work, we chose starvation resistance as a model complex trait because it can be predicted with decent accuracy considering the small sample size [18]. We analyzed the two sexes separately to account for the presence of genetic variation in sexual dimorphism (*i.e.*, cross-sex genetic correlation significantly different from 1) in starvation resistance [22, 24, 25]. We also analyzed startle response, a complex behavioral trait with low genetic variation in sexual dimorphism [22]. Despite the low genetic variation in sexual dimorphism for startle response, we still analyzed the two sexes separately because differences between sexes also exist in the transcriptome [23]. We used line means adjusted for the effect of *Wolbachia* infection and major inversions [21]. After removing lines with missing phenotypic measurements or gene expression profiles, we were left with 198 and 199 lines for starvation resistance and startle response, respectively, for use in all further analyses.

### Transcriptomic prediction

Unless stated, prediction methods assessed in this study follow the general multiple regression model:

$$\mathbf{y} = \mathbf{X}\boldsymbol{\beta} + \mathbf{e} \tag{1}$$

where $\mathbf{y}$ is an $n$-vector of phenotypes for $n$ lines, $\mathbf{X}$ is an $n \times m$ matrix of expression levels for $m$ genes, $\boldsymbol{\beta}$ is an $m$-vector of effect sizes, and $\mathbf{e}$ is an $n$-vector of residuals. We assume that the columns of $\mathbf{X}$ and $\mathbf{y}$ have been centered to mean 0.

**Principal Component Regression (PCR).** PCR [26] uses Principal Component Analysis [27] to reduce the dimensionality of the predictor matrix, $\mathbf{X}$, by selecting a set of $k$ orthogonal components that are linear combinations of the original predictors and maximize their variance. Then, the $n \times k$ matrix of principal components, $\mathbf{T}$, is used in place of $\mathbf{X}$ in Eq 1 [28]. We used the algorithm implemented in the R package pls v. 2.8–2 [29] with default parameters. We used 5-fold cross validation in the training set to select the number of principal components to be used for prediction in the test set.

**Partial Least Squares Regression (PLSR).** Like PCR, PLSR [30] also reduces the dimensionality of the predictor matrix, $\mathbf{X}$. However, this is achieved via a simultaneous decomposition of $\mathbf{X}$ and $\mathbf{y}$ to select a set of $k$ components that maximizes the covariance between $\mathbf{X}$ and $\mathbf{y}$. This addresses a limitation of PCR where the components that best 'explain' $\mathbf{X}$ may not necessarily be the most relevant to $\mathbf{y}$. Then, the $n \times k$ matrix of latent vectors, $\mathbf{T}$, is used in place of $\mathbf{X}$ in Eq 1 [31]. We used the algorithm implemented in the R package pls v. 2.8–2 [29] and chose the 'widekernelpls' method, which is suited for the wide ($m > > n$) matrix

of gene expression, along with other default parameters. We used 5-fold cross validation in the training set to select the number of latent vectors to be used for prediction in the test set. For the startle response analysis, the maximum number of iterations was increased from 100, the default value, to 500 due to complications with model convergence.

**Ridge Regression (RR).** Ridge Regression [7] is a penalized regression method that uses an $\ell_2$ penalty to achieve shrinkage of the estimates of the effect sizes. The amount of shrinkage is determined by a tuning parameter, $\lambda$, such that large values of $\lambda$ result in more shrinkage. We used the algorithm implemented in the R package glmnet v. 4.1–8 [32]. We used 5-fold cross validation to select $\lambda$. All other parameters were left as their default values.

**Least Absolute Shrinkage Selector Operator (LASSO).** LASSO [33] is a penalized regression method that uses an $\ell_1$ penalty to perform both variable selection (by setting some effects to be exactly 0) and shrinkage of the estimates of the effect sizes. The amount of shrinkage and variable selection is determined by a tuning parameter, $\lambda$, such that large values of $\lambda$ result in more shrinkage. We used the algorithm implemented in the R package glmnet v. 4.1–8 [32]. We used 5-fold cross validation to select $\lambda$. All other parameters were left as their default values.

**BayesC.** BayesC [9] is a Bayesian regression method that imposes a spike-and-slab prior on the effect sizes:

$$\beta_j \sim \pi N(0, \sigma_\beta^2) + (1 - \pi)\delta_0 \tag{2}$$

where $\pi$ is the probability that the effect of the $j$th variable comes from a Normal distribution with mean 0 and variance $\sigma_\beta^2$, and $\delta_0$ is a point-mass at 0. In this way, both variable selection and effect shrinkage are achieved. In the R package BGLR v. 1.1.0 [34] implementation, the posterior distribution of the effect sizes and some model parameters are computed using Markov Chain Monte Carlo (MCMC) methods. We ran the algorithm for 130,000 iterations, discarded the first 30,000 samples as burn-in, and retained every 50th sample. We assessed convergence through visual inspection of the trace plots. The expected proportion of variance explained by the predictors, $R^2$, was set to 0.8 in accordance with the broad sense heritability estimates of line means for starvation resistance and startle response [22, 25].

**Variational Bayesian Variable Selection (VARBVS).** VARBVS is a Bayesian regression method that imposes the same spike-and-slab prior as BayesC. However, VARBVS computes posterior distributions using Variational Inference, which is computationally more efficient than MCMC [35]. The algorithm implemented in the R package varbvs v. 2.6–10 [35] was fit with default parameters.

**Multiple Regression with Adaptive Shrinkage (MR.ASH).** MR.ASH is a Bayesian regression method that imposes a scale mixture-of-Normals prior on the effect sizes:

$$\beta_j | \boldsymbol{\pi}, \boldsymbol{\sigma^2} \sim \pi_0 \delta_0 + \sum_{k=1}^{K} \pi_k N(0, \sigma_k^2) \tag{3}$$

for a fixed grid of variances, $\boldsymbol{\sigma^2}$. Thus, like BayesC and VARBVS, MR.ASH performs both variable selection and effect shrinkage. MR.ASH is able to model complex distributions of the effect sizes thanks to a more flexible prior than VARBVS and BayesC. MR.ASH uses a Variational Empirical Bayes approach to estimate the prior (*i.e.*, the mixture weights, $\boldsymbol{\pi}$) from the data and compute the posterior distribution of the effect sizes [36]. The algorithm implemented in the R package mr.ash.alpha v. 0.1–43 [37] was fit with default parameters and initialized using the effect size estimates from LASSO.

**Transcriptomic Best Linear Unbiased Predictor (TBLUP).** TBLUP [38] is a linear mixed model that aggregates the effects of all of the genes into a single random effect. Let

$\mathbf{t} = \sum_{j=1}^{m} \mathbf{w}_j \beta_j = \mathbf{W}\boldsymbol{\beta}$, where $\mathbf{w}_j$ is a standardized version of $\mathbf{x}_j$ to have unit variance, then:

$$\mathbf{y} = \mathbf{t} + \mathbf{e} \qquad (4)$$

where $\mathbf{t}$ is a $n$-vector of transcriptomic effects, $\mathbf{t} \sim N(0, \mathbf{T}\sigma_{\mathbf{t}}^2)$, and $\mathbf{T} = \mathbf{T} = \dfrac{\mathbf{W}\mathbf{W}^{\mathsf{T}}}{m}$ is the Transcriptomic Relationship Matrix (TRM). TBLUP was implemented by using the R package BGLR v. 1.1.0 [34]. BGLR uses a Bayesian approach to estimate the transcriptomic and residual variance components. We ran the algorithm for 85,000 iterations, discarded the first 10,000 samples as burn-in, and retained every 50$^{\text{th}}$ sample. We assessed convergence through visual inspection of the trace plots. The expected proportion of variance explained by the predictors, $R^2$, was set to 0.8 in accordance with the broad sense heritability estimates of line means for starvation resistance and startle response [22, 25]. All other parameters were left as their default values.

All of the previous methods assume that no gene-gene interactions affect the phenotype. Thus, we decided to add some more flexible machine learning methods capable of capturing interaction effects to the comparison.

**Random Forest (RF).**   Random Forest is a machine learning method whereby a collection of decision trees are grown, each on a different bootstrap sample of the predictor data [11]. This method has been used to identify gene-gene interactions successfully [39]. The model is given by:

$$\mathbf{y} = \sum_{s=1}^{S} c_s h_s(\mathbf{y}; \mathbf{X}) \qquad (5)$$

where $S$ is the number of decision trees, $c_s$ is a shrinkage factor that averages the trees, $h_s(\mathbf{y};\mathbf{X})$ is a decision tree that is grown using only a subset of predictors at each node [11]. The algorithm implemented in the R package partykit v. 1.2–20 [40] was fit with 1000 trees and default parameters.

**Neural Networks (NN).**   Artificial Neural Networks are a type of machine learning method that use layers of nodes, or neurons, to process data similar to how the human brain works. Neural networks are built using input layers, hidden layers, and output layers [41]. Networks can vary by hidden layer count, neuron count per layer, and activation function per layer. Neuron count selection is a fundamental problem in constructing networks [42]. Neural networks use nonlinear activation functions to determine whether neurons in hidden layers should be activated based on their inputs. This feature can be used to model gene-gene interactions [43]. The neural network implemented through the R package neuralnet v1.44.2 [44] used default parameters and a custom neuron structure with a hidden layer of 1,000 neurons. In our model, weights for each neuron in the hidden layer were learned using resilient back-propagation [45].

## Gene Ontology-informed transcriptomic prediction

While some of the methods above try to enrich the prediction model for genes that are particularly predictive of the trait by performing internal variable selection, this procedure becomes difficult with a small sample size such as in the DGRP. Informing prediction models with functional annotation has been shown to be effective at disentangling signal from noise and improve accuracy in complex trait prediction [18, 46–48]. Edwards *et al.* [46] and Morgante *et al.* [18] used Gene Ontology (GO) annotations [49] to improve prediction accuracy for three complex traits in *Drosophila*. However, these applications only used BLUP-type models to include GO information. Here, we tested two additional methods described below. For each

sex, we selected GO terms that included at least five genes present in the DGRP expression data, in line with previous work [18]. This procedure resulted in 2,628 terms for females and 2,580 terms for males being retained for further analysis. For all methods, GO-informed models were fit with one GO term at a time for all GO terms specified for each sex.

**Sparse Group LASSO.** Sparse group LASSO [50] is a penalized regression method that uses a combination of the $\ell_1$ penalty and a group LASSO penalty [50]. The Group LASSO [51] applies variable selection on entire groups of predictors, while the $\ell_1$ penalty achieves effect shrinkage and variable selection at the individual variable level. The strength of the penalties is determined by a tuning parameter, $\lambda$, such that larger values of $\lambda$ result in more shrinkage/selection. In our application, one group included all of the genes in the selected GO term and the other group included all of the remaining genes. We used the Sparse Group LASSO implementation in the R package sparseGL v1.0.2 [52] with default parameters.

**GO-BayesC.** GO-BayesC is an extension of BayesC that imposes independent spike-and-slab priors on the effect sizes of genes grouped by GO term association. It follows the model

$$\mathbf{y} = \mathbf{X}_{GO}\boldsymbol{\beta}_{GO} + \mathbf{X}_{notGO}\boldsymbol{\beta}_{notGO} + \mathbf{e} \tag{6}$$

where $\mathbf{X}_{GO}$ is the subset of $\mathbf{X}$ containing the genes associated with the selected GO term, $\boldsymbol{\beta}_{GO}$ is the vector of effects of the genes in the selected GO term, $\mathbf{X}_{notGO}$ is the subset of $\mathbf{X}$ containing all other genes, and $\boldsymbol{\beta}_{notGO}$ is the vector of effects of all other genes. $\beta_{GO,j}$ and $\beta_{notGO,j}$ are assigned separate spike-and-slab priors as in Eq 2, with group-specific prior inclusion probability and effect size variance. This method uses the same algorithm from the R package BGLR [34]. We ran the algorithm for 130,000 iterations, discarded the first 30,000 samples as burn-in, and retained every 50$^{th}$ sample. We assessed convergence through visual inspection of the trace plots. The expected proportion of variance explained by the predictors, $R^2$, was set to 0.8 in accordance with the broad sense heritability estimates of line means for starvation resistance and startle response [22, 25].

**GO-TBLUP.** GO-TBLUP is an extension of TBLUP that includes two random effects—one associated with genes in the selected GO term and one associated with all of the other genes. It follows the model

$$\mathbf{y} = \mathbf{t}_{GO} + \mathbf{t}_{notGO} + \mathbf{e} \tag{7}$$

where $\mathbf{t}_{GO}$ is a $n$-vector of transcriptomic effects associated with genes in the GO term, $\mathbf{t}_{GO} \sim N(0, \mathbf{T}_{GO}\sigma^2_{\mathbf{t}_{GO}})$, $\mathbf{T}_{GO} = \frac{\mathbf{W}_{GO}\mathbf{W}^{\mathsf{T}}_{GO}}{m}$, $\mathbf{W}_{GO}$ is the subset of $\mathbf{W}$ containing the genes associated with the selected GO term, $\mathbf{t}_{notGO}$ is a $n$-vector of transcriptomic effects associated with all other genes, $\mathbf{t}_{notGO} \sim N(0, \mathbf{T}_{notGO}\sigma^2_{\mathbf{t}_{notGO}})$, $\mathbf{T}_{notGO} = \frac{\mathbf{W}_{notGO}\mathbf{W}^{\mathsf{T}}_{notGO}}{m}$, and $\mathbf{W}_{notGO}$ is the subset of $\mathbf{W}$ containing all other genes. GO-TBLUP was implemented by using the R package BGLR v. 1.1.0 [34]. BGLR uses a Bayesian approach to estimate the transcriptomic and residual variance components. We ran the algorithm for 85,000 iterations, discarded the first 10,000 samples as burn-in, and retained every 50$^{th}$ sample. We assessed convergence through visual inspection of the trace plots. The expected proportion of variance explained by the predictors, $R^2$, was set to 0.8 in accordance with the broad sense heritability estimates of line means for starvation resistance and startle response [22, 25]. All other parameters were left as their default values.

## Evaluation scheme

We fitted each method to 90% of the data (*i.e.*, the training set) to estimate the model parameters. The trained model was then validated by predicting phenotypes for the remaining 10% of

the data (*i.e.*, the test set). Prediction accuracy was measured as the correlation coefficient between the observed and predicted phenotypes. We repeated this procedure for 25 random training-test splits and used the average correlation across splits as our final metric to assess prediction accuracy.

## Results

The figures and tables for the analyses of starvation resistance are reported in the main text, while those for startle response can be found in the supplementary materials.

### Transcriptomic prediction

We first fitted a few widely used prediction methods and compared their accuracy. The results for starvation resistance are shown in Fig 1 and S1 Table. Overall, prediction accuracy was low to moderate for all methods, especially considering that the analyses were based on lines means of many individual flies, which substantially increases the broad sense heritability of the trait to values around 0.8 (the broad sense heritability of line means is $H_m^2 = \frac{\sigma_G^2}{\sigma_G^2 + \frac{\sigma_E^2}{f}}$, where $\sigma_G^2$ is the genetic variance, $\sigma_E^2$ is the environmental variance, and $f$ is the number of measured flies per line) [22, 53]. However, differences in prediction accuracy between methods existed, both

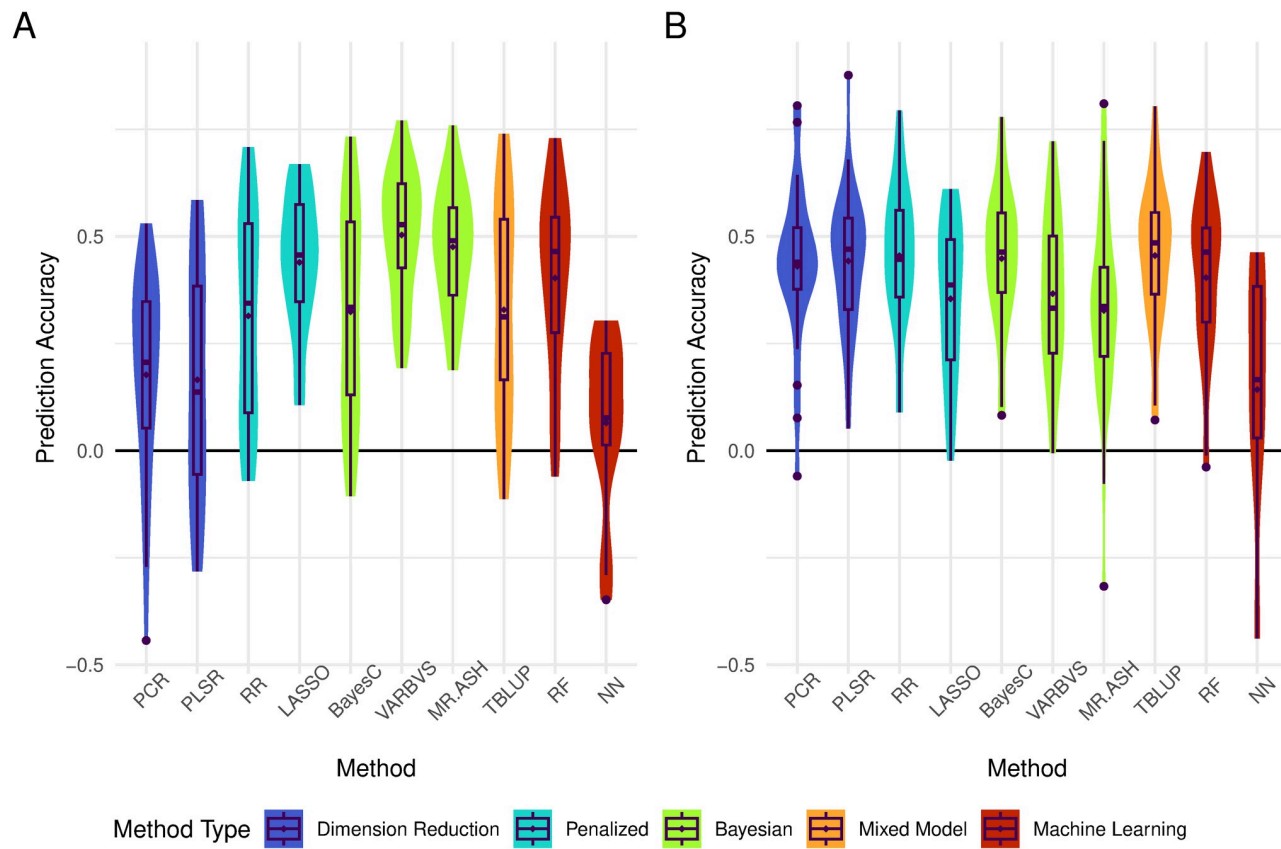

**Fig 1. Prediction accuracy for starvation resistance.** Prediction accuracy of 25 replicates in females (A) and males (B) for all standard methods. Methods are colored by family, where dimension reduction (blue), penalized regression (cyan), Bayesian regression (lime), linear mixed model (orange), and machine learning methods (red) are ordered from left to right. The mean correlation coefficient is denoted by diamonds. Outliers are denoted by circles.

within each sex and across sexes. In males, we found that TBLUP ($r = 0.455\pm0.035$) and Ridge Regression ($r = 0.455\pm0.034$) provided the highest accuracy, with BayesC ($r = 0.448\pm0.033$), PLSR ($r = 0.442\pm0.034$), and PCR ($r = 0.430\pm0.039$) being competitive. On the other hand, Neural Network provided the lowest prediction accuracy for males ($r = 0.143\pm0.079$). In females, we observed more marked differences in prediction accuracy across methods. Methods that perform variable selection—*i.e.*, VARBVS, MR.ASH, and LASSO—tended to perform better than the other methods, with VARBVS ($r = 0.503\pm0.031$) providing the highest accuracy. Neural Network provided the lowest prediction accuracy in females ($r = 0.064\pm0.051$) as well.

The results for startle response are presented in S1 Fig and S2 Table. A few methods (*i.e.*, PCR, PLSR, RR, LASSO, and NN) produced intercept-only models or failed to converge, resulting in no correlation coefficient to measure prediction accuracy for some replicates (S3 Table). In particular, LASSO produced intercept-only models for 23 (92%) and 12 (48%) replicates in females and males, respectively. PLSR failed to converge for 20 (80%) replicates in females. Thus, we excluded LASSO and PLSR from our comparisons as both did not have accuracy for more than 40% of replicates in at least one sex, making the results unreliable. Of the remaining methods, the best performing ones are BayesC ($r = 0.182\pm0.045$ for females, $r = 0.273\pm0.041$ for males) and TBLUP ($r = 0.173\pm0.045$ for females, $r = 0.258\pm0.042$ for males) in both sexes. In females, all of the other methods achieved a correlation lower than 0.100, with PCR performing the worst ($r = -0.025 \pm 0.069$; 15 replicates). In males, RF ($r = 0.251 \pm 0.047$) and MR.ASH ($r = 0.236 \pm 0.049$) achieved prediction accuracies comparable to the top methods. All of the remaining methods achieved a correlation lower than 0.200, with Neural Network achieving the lowest prediction accuracy ($r = -0.028 \pm 0.053$; 15 replicates).

Collectively, these results show that differences in prediction accuracy between methods are present, with performance being dependent on the genetic architecture of the trait and the methods' assumptions.

## Gene Ontology informed transcriptomic prediction

It has been shown previously that informing prediction models with functional information can improve prediction accuracy [18, 46–48]. Thus, we also tested methods that could include external information. In this work, we focused on GO annotation and extensions of BayesC and TBLUP, namely GO-BayesC and GO-TBLUP. The results are summarized in Fig 2 and S4 Table for starvation resistance, and in S2 Fig and S5 Table for startle response. We also sought to use the Sparse Group LASSO. However, in our initial testing using starvation resistance, the prediction accuracies provided by that method were nearly identical for all GO terms tested (S3 Fig). This pattern was also seen for GO terms found to be highly predictive by GO-BayesC and GO-TBLUP. Thus, we decided not to assess Sparse Group LASSO further.

For both traits analyzed, we found that GO-BayesC and GO-TBLUP provided accuracies that were similar to or lower than the respective standard model (*i.e.*, BayesC and TBLUP) for the majority of GO terms in both sexes. However, some GO terms seemed to be particularly predictive of the trait, yielding accuracies that were substantially higher than the standard models. For starvation resistance, the accuracies provided by GO-BayesC and GO-TBLUP generally agreed for both sexes ($r = 0.836$), as shown in Fig 3. The same pattern held for startle response in both females ($r = 0.856$) and males ($r = 0.872$), as shown in S4 Fig.

In females, four of the five most predictive GO terms are shared between GO-BayesC and GO-TBLUP for starvation resistance. GO:0017056 (GO-BayesC $r = 0.477 \pm 0.034$, GO-TBLUP $r = 0.457 \pm 0.041$) and GO:0006606 (GO-BayesC $r = 0.464 \pm 0.039$, GO-TBLUP

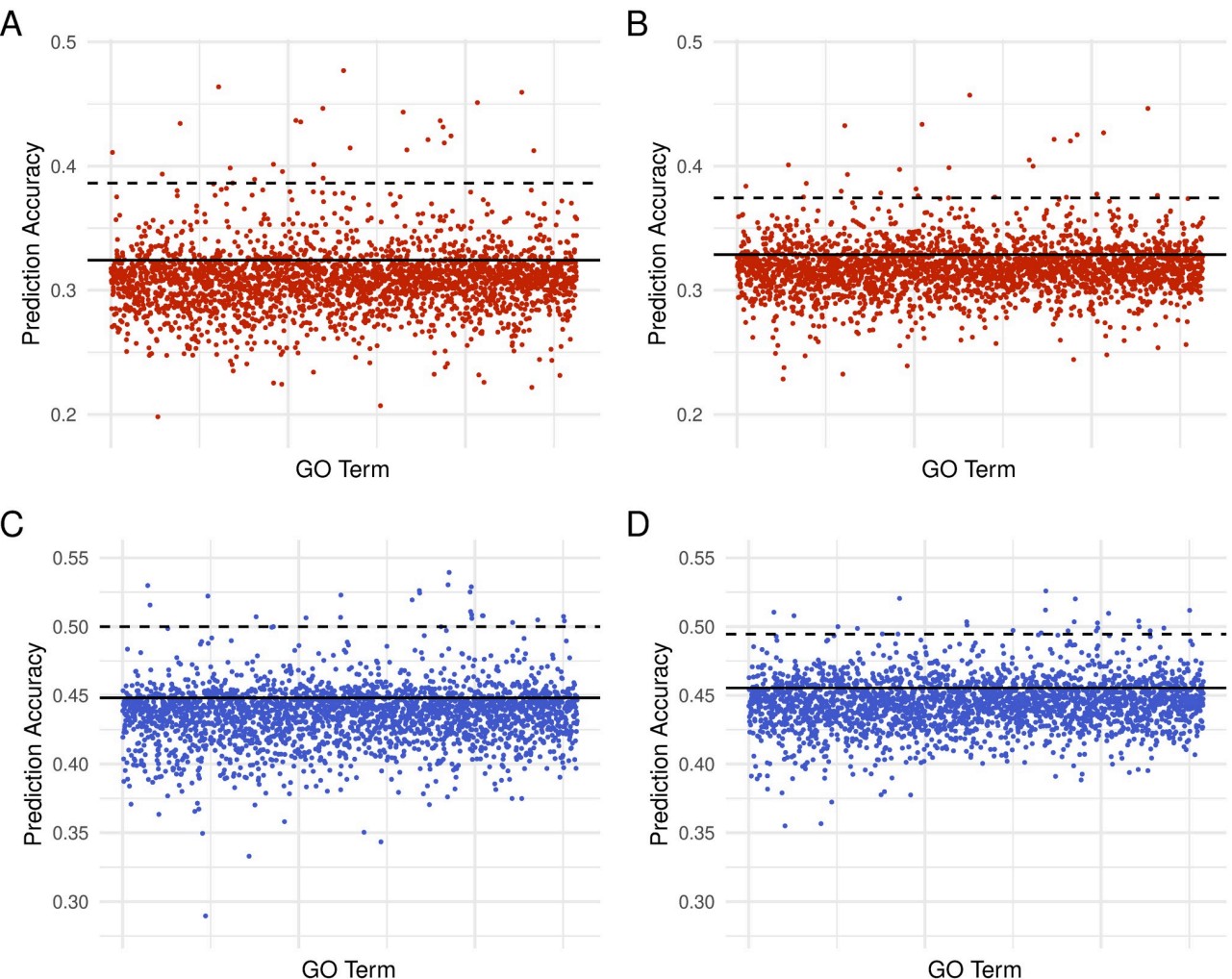

**Fig 2. Prediction accuracy for starvation resistance using GO terms.** Prediction accuracy in the two sexes using GO-BayesC (A for females, C for males) and GO-TBLUP (B for females, D for males). Each dot represents the mean correlation between true and predicted phenotypes (*r*) across 25 replicates for a GO term. The solid line indicates the mean *r* from the respective standard method (*i.e.*, BayesC and TBLUP). The dashed black line represents the 99th percentile of terms ranked by prediction accuracy.

*r* = 0.434 ± 0.044) are both related to nuclear import by function and structure, respectively. Lee *et al.* has demonstrated that starvation resistance induces nuclear pore degradation in yeast [54]. GO:0055088 (GO-BayesC *r* = 0.446 ± 0.040, GO-TBLUP *r* = 0.460 ± 0.036) and GO:0045819 (GO-BayesC *r* = 0.451 ± 0.038, GO-TBLUP *r* = 0.427 ± 0.042) are related to macromolecule metabolism in lipids and carbohydrates, respectively. GO:0055088 has been implicated in starvation resistance using Korean rockfish [55]. GO:0017056 was the most predictive term for GO-BayesC (*r* = 0.477 ± 0.034) and GO-TBLUP (*r* = 0.457 ± 0.041). However, some differences between methods existed. For example, GO:0016042, which is involved in lipid catabolism, was found to be highly predictive by GO-BayesC (*r* = 0.447 ± 0.038), while GO:0008586, which is involved in wing vein morphogenesis, was highly predictive in GO-TBLUP (*r* = 0.424 ± 0.039). For startle response, GO:0008061 (GO-BayesC *r* = 0.294 ± 0.043, GO-TBLUP *r* = 0.262 ± 0.046) and GO:0043066 (GO-BayesC *r* = 0.303 ± 0.042, GO-TBLUP *r* = 0.270 ± 0.047) were found within the top five most predictive

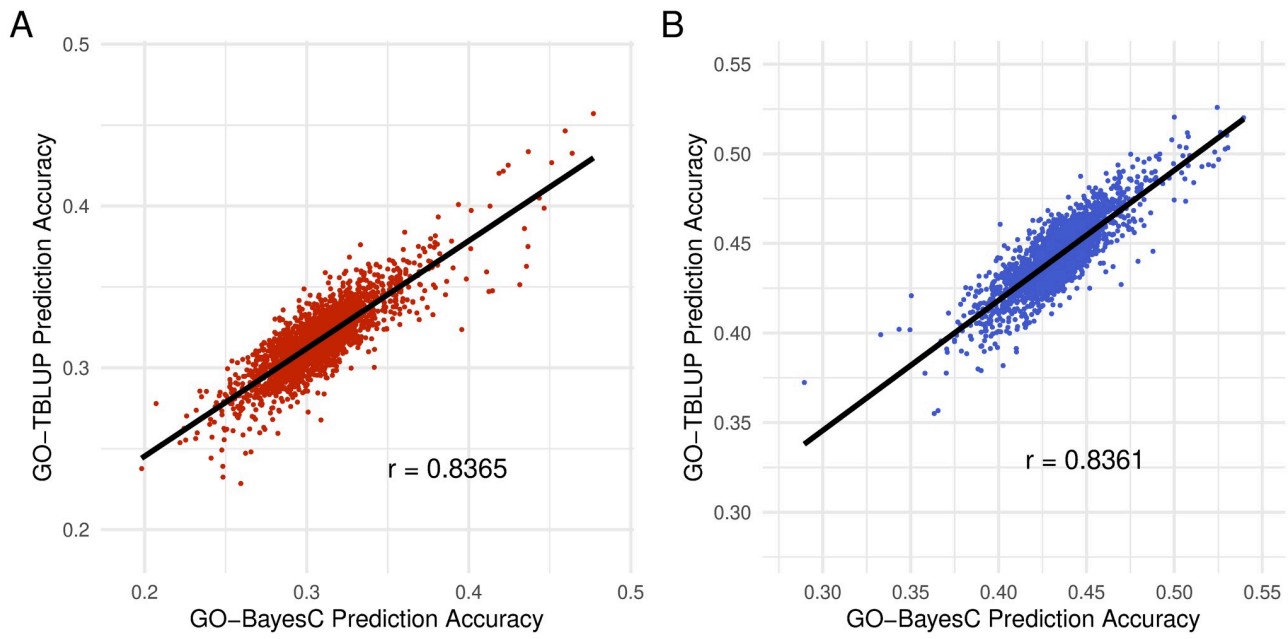

**Fig 3. Correlation of prediction accuracy of GO-annotated methods for starvation resistance.** Prediction accuracy for all GO terms of GO-BayesC (x-axis) against GO-TBLUP (y-axis) for females (A) and males (B). The black line represents the line of least squares fit for each panel.

GO terms in both methods for females. These terms cover chitin binding (GO:0008061) and negative regulation of apoptotic processes (GO:0043066).

In males, two of the five most predictive GO terms are shared between methods for starvation resistance. GO:0042593 (GO-BayesC $r = 0.539 \pm 0.032$, GO-TBLUP $r = 0.520 \pm 0.035$) is involved in glucose homeostasis, while GO:0035003 (GO-BayesC $r = 0.526 \pm 0.036$, GO-TBLUP $r = 0.512 \pm 0.030$) is involved in the subapical complex. This is involved with nutrient acquisition in the intestines as part of the barrier between host cells and the gut microbiome [56]. Four of the top five GO terms found by either method are implicated in cellular growth and development. In GO-BayesC, GO:0042461 ($r = 0.530 \pm 0.036$) is involved in photoreceptor cells, GO:0001738 ($r = 0.530 \pm 0.036$) is involved in epithelial tissue and GO:0045186 ($r = 0.529 \pm 0.037$) is involved in the assembly of the zonula adherens. GO:0007485, which is involved in genital disc formation, was highly predictive in GO-TBLUP ($r = 0.520 \pm 0.030$). Multiple studies have found connections between cell size regulation and overall body size with starvation resistance [57–59]. The top GO term for TBLUP in males, GO:0035008 ($r = 0.526 \pm 0.026$), is involved in the positive regulation of the melanization defense response. The biological connection of this process to starvation resistance is unclear, as this response increases oxidative stress in wounds to prevent infection [60]. For startle response, four of the top five most predictive GO terms are shared between methods in males. These terms include chitin binding (GO:0008061—GO-BayesC $r = 0.294 \pm 0.043$, GO-TBLUP $r = 0.262 \pm 0.046$), imaginal disc growth factor receptor binding (GO:0008084—GO-BayesC $r = 0.368 \pm 0.031$, GO-TBLUP $r = 0.365 \pm 0.028$), protein tyrosine phosphatase activity (GO:0004725—GO-BayesC $r = 0.348 \pm 0.032$, GO-TBLUP $r = 0.349 \pm 0.033$), and plasma membrane (GO:0005886—GO-BayesC $r = 0.348 \pm 0.037$, GO-TBLUP $r = 0.358 \pm 0.035$).

Overall, the increased prediction accuracy for both traits provided by specific GO terms highlights the usefulness of external information for improving accuracy.

## Gene analysis

Given that many of the most predictive GO terms were biologically relevant to the traits analyzed, we investigated whether any particular genes were included in such terms. For starvation resistance, we selected the 1% most predictive GO terms for each method and sex (26 and 25 GO terms for females and males, respectively). For each prediction method and sex combination, we counted how many times each gene was found across these GO terms. We then examined the distribution of the count (S5 Fig) and decided to focus only on the most frequently occurring genes. This resulted in selecting genes appearing in 5 or more GO terms for GO-TBLUP in females and genes appearing in 4 or more GO terms for all other method and sex combinations. These results are summarized in Table 1 and S6 Table. For startle response, a similar procedure led us to selecting genes appearing in 4 or more top GO terms for all method and sex combinations (S6 Fig). These results are summarized in S7 and S8 Tables.

For both sexes, the results show that some overlapping genes were found by both GO-BayesC and GO-TBLUP, while some genes were picked up by only one method. For starvation resistance, significant enrichment (Fisher's Exact Test $P < 0.001$) of protein kinases emerged when considering top genes across all setups. For startle response, top genes were enriched (Fisher's Exact Test $P < 0.001$) for neuronal development and sensory receptors across all setups.

In females, GO-BayesC and GO-TBLUP found *AkhR*, *Akt1*, *InR*, *Egfr*, and *Erk7* in common out of the top 1% of GO terms for starvation resistance. Of these genes, *AkhR*, *Akt1*, and *InR* are related to insulin signaling and lipid metabolism [61–63]. Aside from insulin signaling, *Egfr* and *Erk7* have been implicated in starvation resistance [64, 65]. For startle response, only *crb*, a gene involved in *Notch* regulation and photoreceptor morphogenesis [66], was shared between GO-BayesC and GO-TBLUP. For starvation resistance, most of the genes found by only GO-BayesC are from the nucleoporin family (*mbo*, *Nup53*, *Nup54*, *Nup93–1*, *Nup93–2*,

**Table 1. Most frequent genes across the top 1% of GO terms for starvation resistance.**

| GO-BayesC Female | | GO-TBLUP Female | | GO-BayesC Male | | GO-TBLUP Male | |
|---|---|---|---|---|---|---|---|
| Gene | Count | Gene | Count | Gene | Count | Gene | Count |
| AkhR | 9 | Egfr | 8 | sdt | 13 | sdt | 8 |
| mbo | 6 | Akt1 | 6 | aPKC | 9 | aPKC | 7 |
| InR | 6 | AkhR | 6 | par-6 | 8 | PDZ-GEF | 6 |
| Nup54 | 5 | InR | 6 | Patj | 8 | par-6 | 6 |
| Akh | 5 | Sik3 | 5 | scrib | 5 | Patj | 5 |
| Nup154 | 4 | emc | 5 | baz | 5 | crb | 5 |
| Nup93–1 | 4 | put | 5 | crb | 5 | Ilp2 | 4 |
| Nup205 | 4 | babo | 5 | Moe | 5 | Desat1 | 4 |
| Nup93–2 | 4 | Pdk1 | 5 | PDZ-GEF | 5 | InR | 4 |
| Nup98–96 | 4 | Ack | 5 | AkhR | 4 | Ras85D | 4 |
| Nup153 | 4 | Pkc98E | 5 | Ilp2 | 4 | | |
| Akt1 | 4 | CG3216 | 5 | InR | 4 | | |
| Pi3K92E | 4 | CG31183 | 5 | dlg1 | 4 | | |
| Egfr | 4 | Erk7 | 5 | shg | 4 | | |
| Erk7 | 4 | hpo | 5 | | | | |
| | | hep | 5 | | | | |

Top overlapping genes across the 1% most predictive GO terms for each method and sex combination. Genes appearing in all four setups are highlighted in green. Genes appearing in both methods for females are highlighted in red. Genes appearing for both methods in males are highlighted in blue.

*Nup98–96*, *Nup153*, and *Nup205*). The remaining genes are tangential to the *InR* signaling pathway. For startle response, *aPKC* was the only other gene found by GO-BayesC. For starvation resistance, eleven distinct genes found by GO-TBLUP only are involved in various complexes and pathways. *hpo*, *Ack*, and *Sik3* are related to the Hippo or Salvador-Warts-Hippo signaling pathway [67–69]. *pdk1* is part of the *InR* signaling pathway [70]. *put* and *babo* are involved in the activin signaling pathway [71]. *hep* is a kinase involved in regulation of gut metabolism [72]. For startle response, *Orco*, *dco*, and *park* were found by GO-TBLUP only. These genes are related to nervous system functions, where *Orco* is involved in olfactory receptors [73], *dco* is part of circadian signaling [74], and *park* is related to locomotion [75].

In males, GO-BayesC and GO-TBLUP found *sdt*, *aPKC*, *PDZ-GEF*, *par-6*, *Patj*, *crb*, *Ilp2*, and *InR* in common out of the top 1% of GO terms for starvation resistance. The two major categories that emerge from these genes are carbohydrate metabolism and cell polarization. For carbohydrate metabolism, the insulin-like receptor *InR* and an insulin-like peptide *Ilp2* are key components of *InR* signaling [76]. *InR* is the only gene that was found by both GO methods in both sexes. The remaining genes are all related to cell polarization. From both methods, *crb*, *sdt*, and *Patj* are involved in the Crumbs complex [66], while *aPKC* and *par6* are involved in the PAR complex [77]. Outside of these complexes, *PDZ-GEF* is involved in epithelial cell polarization [78]. The startle response analysis yielded two prevalent genes in common between GO-BayesC and GO-TBLUP. *Idgf2* is involved in stress response [79], while *stg* is involved in cell cycle progression [80]. Six genes were found by GO-BayesC only in the top 1% of GO terms in males for starvation resistance. *baz*, *scrib*, and *dlg1* are related to the Scribble and PAR complexes [81], while *Moe* regulates the Crumbs complex [82]. *AkhR*, described above, and *shg*, which is part of the *Egfr* signaling pathway [83], were also found uniquely by GO-BayesC. For startle response, GO-BayesC found five *Idgf* genes (in addition to *Idgf2*) and neuromuscular genes *wg*, *Ptp99A*, and *Ptp69D*. Only two genes were found uniquely by GO-TBLUP in males for starvation resistance. *Desat1* is a lipid desaturase [84], while *Ras85D* is an oncogenic cell growth promoter [85]. For startle response, GO-TBLUP uniquely found *Notch*, *Cul3*, and *Dl*.

Detailed descriptions of the genes and their connections with starvation resistance and startle response are given in S1 Text.

## Discussion

In this study, we evaluated ten statistical methods on their ability to predict starvation resistance and startle response, two well-documented quantitative traits [22], using transcriptomic data [23]. As expected, we found differences in the prediction accuracy provided by the methods tested, both within each sex and between sexes. While most methods were somewhat predictive, neural networks provided minimal prediction accuracy for both sexes. This is in agreement with previous work showing the importance of feature selection prior to model fitting in the $p >> n$ regime for neural networks to perform well [12]. However, we caution the readers that our work focused on out-of-the-box performance of the different methods. Better performance may be achieved with additional fine-tuning of each method. For starvation resistance, the most predictive methods in females, VARBVS and MR.ASH, are both Bayesian regression methods that perform effect shrinkage and variable selection. These methods allow for the underlying genetic architecture to be sparse, suggesting that not all genes affect starvation resistance. In contrast, the most predictive methods in males (TBLUP and ridge regression) only perform effect shrinkage, which suggests that the genetic architecture of starvation resistance is denser in males. The difference in best performing methods between sexes is not

surprising because starvation resistance is known to have high genetic variation in sexual dimorphism, *i.e.*, the two sexes have different genetic architectures [22, 24, 25].

The results for startle response, a complex behavioral trait with low genetic variation in sexual dimorphism, show that prediction accuracy for all methods is substantially lower than for starvation resistance and is higher in males than in females. However, we note that a few methods (*i.e.*, PCR, PLSR, RR, LASSO, and NN) could not find signal in the data or failed to converge for several replicates. While this is a result in itself in that these methods seem less robust, it also limits the reliability of the comparisons. Among the methods that did not have any issues, BayesC and TBLUP achieved the highest prediction accuracies for both sexes, in agreement with the low genetic variation in sexual dimorphism for startle response.

Overall, the results for both traits highlight the importance of choosing methods with assumptions that match the genetic architecture of the trait under investigation. Additionally, prediction analysis can also provide some hypotheses about the genetic architecture of the traits, which guide further specific experiments and analyses.

Previous studies have shown that including functional annotation information into prediction models can help improve prediction accuracy [18, 46–48]. Thus, we selected two methods that allow the incorporation of additional information, BayesC and TBLUP, and annotated them with Gene Ontology (GO) information. We showed that a small number of biologically relevant GO terms achieved substantially higher prediction accuracies for GO-BayesC and GO-TBLUP than the standard BayesC and TBLUP methods. While the correlation between prediction accuracies for each GO between GO-BayesC and GO-TBLUP was high for both sexes ($r = 0.84 - 0.87$), differences in method assumptions resulted in different top GO terms between methods, especially in males. For starvation resistance, the most predictive GO terms for both sexes shared genes with biological connections with the trait (S1 Text). For example, *InR* and *AkhR* are involved in carbohydrate and lipid metabolism. Both genes have been associated to starvation resistance in previous studies [86, 87]. However, many of the genes shared by the most predictive GO terms in the two sexes were different (*e.g.*, *Egfr* in females and *sdt* in males). These findings also suggest that both methods may be able to highlight sex-shared and sex-specific genes of interest for traits with unknown genetic architectures. The results for startle response showed similar trends. A few GO terms substantially increased prediction accuracy, with the most predictive GO term (*e.g.*, GO:0008061—chitin binding) suggesting a plausible biological relation to startle response through neurological mechanisms [88]. The most predictive GO terms also included genes related to startle response mechanisms (*e.g.*, *crb* regulating sensory receptors in females, and *Notch* and *Dl* promoting neuron development in males [82, 89]). Overall, our findings suggest that additional information from GO terms may help disentangle signal from noise to improve prediction accuracy and our understanding of the complex trait of interest.

In conclusion, we found that differences in prediction performance between methods exist and depend on the assumptions made by the model relative to the genetic architecture of the trait of interest. We also confirmed that external information, such as GO term annotation, can improve prediction accuracy for biologically relevant data. However, there are a number of limitations and considerations to address. First, the data is limited by the small number of available DGRP lines. In fact, the estimates of the effects of the $p = 12,000$ genes using only $n = 200$ lines may not be precise, resulting in low prediction accuracy. We hypothesize that increasing the sample size of the DGRP would improve signal detection and estimation for all methods and traits. Second, the linear regression performed by most methods used here do not account for interactions between genes and may perform poorly for complex traits with epistatic interactions. At the same time, accounting for interactions results in many more effects to be estimated, which suffers from small sample sizes, such as

the DGRP, even more. Previous work describing a deep neural network architecture for detecting gene-gene interactions notes that these types of methods are generally applicable to larger data sets [43]. The poor performance of neural networks in our study also seems to confirm this issue. Third, the gene expression profiles were obtained from whole flies reared under standard conditions. Higher prediction accuracy may be achieved if gene expression were measured under the same conditions used to score starvation resistance and startle response. For similar reasons, higher prediction accuracy may also be achieved when using gene expression from only tissues relevant to the trait analyzed (*e.g.*, brain tissues for startle response). Despite these limitations, we have shown that gene expression data coupled with appropriate model selection and external information can be effective for complex trait prediction.

## Supporting information

**S1 Fig. Prediction accuracy for startle response.** Prediction accuracy for 25 replicates in females (A) and males (B) for all standard methods. Methods are colored by family, where dimension reduction (blue), penalized regression (cyan), Bayesian regression (lime), linear mixed model (orange), and machine learning methods (red) are ordered from left to right. The mean correlation coefficient is denoted by diamonds. Outliers are denoted by circles.
(TIF)

**S2 Fig. Prediction accuracy for startle response using GO terms.** Prediction accuracy in the two sexes using GO-BayesC (A for females, C for males) and GO-TBLUP (B for females, D for males). Each dot represents the mean correlation between true and predicted phenotypes (*r*) across 25 replicates for a GO term. The solid line indicates the mean *r* from the respective standard method (*i.e.*, BayesC and TBLUP). The dashed black line represents the 99[th] percentile of terms ranked by prediction accuracy.
(TIF)

**S3 Fig. Prediction accuracy using Sparse Group LASSO.** Violin plot comparison of Sparse Group Lasso results for top GO terms from GO-BayesC/GO-TBLUP along with randomly selected GO terms in females.
(TIF)

**S4 Fig. Correlation of prediction accuracy of GO-annotated methods for startle response.** Prediction accuracy for all GO terms using GO-BayesC (x-axis) against GO-TBLUP (y-axis) for females (A) and males (B). The black line represents the line of least squares fit for each panel.
(TIF)

**S5 Fig. Histogram of genes present in top GO terms for starvation resistance.** Distribution of number of overlapping genes for the top 1% of GO terms in the two sexes using GO-BayesC (A for females, C for males) and GO-TBLUP (B for females, D for males). The selection cutoff is marked by the vertical bar.
(TIF)

**S6 Fig. Histogram of genes present in top GO terms for startle response.** Distribution of number of overlapping genes for the top 1% of GO terms in the two sexes using GO-BayesC (A for females, C for males) and GO-TBLUP (B for females, D for males). The selection cutoff is marked by the vertical bar.
(TIF)

**S1 Table. Prediction accuracy for starvation resistance.** Mean prediction accuracy and standard error for all methods in females and males.
(XLSX)

**S2 Table. Prediction accuracy for startle response.** Mean prediction accuracy and standard error for all methods in females and males.
(XLSX)

**S3 Table. Successful replicates for startle response.** Number of successful replicates out of 25 replicates for all methods in both sexes.
(XLSX)

**S4 Table. Prediction accuracy for starvation resistance for each GO term.** Mean prediction accuracy and standard error for each GO term in GO-BayesC and GO-TBLUP for females and males.
(XLSX)

**S5 Table. Prediction accuracy for startle response for each GO term.** Mean prediction accuracy and standard error for each GO term in GO-BayesC and GO-TBLUP for females and males.
(XLSX)

**S6 Table. All genes across top GO terms for starvation resistance.** Genes in the top 1% of GO terms for GO-BayesC and GO-TBLUP in females and males ordered by gene count.
(XLSX)

**S7 Table. Most frequent genes across the top 1% of GO terms for startle response.** Top overlapping genes across the 1% most predictive GO terms for each method and sex combination. Genes appearing in all four setups are highlighted in green. Genes appearing in both methods for females are highlighted in red. Genes appearing for both methods in males are highlighted in blue.
(XLSX)

**S8 Table. All genes across top GO terms for startle response.** Genes in the top 1% of GO terms for GO-BayesC and GO-TBLUP in females and males ordered by gene count.
(XLSX)

**S1 Text. Details about the genes underlying the most predictive GO terms.** Detailed description of the genes identified in the gene analysis, including their relevance to the traits of interest.
(PDF)

## Acknowledgments

We thank Trudy Mackay for helpful comments on an earlier version of this manuscript, and Liangjiang Wang for suggestions about the neural network analyses.

## Author Contributions

**Conceptualization:** Fabio Morgante.

**Data curation:** Noah Klimkowski Arango.

**Formal analysis:** Noah Klimkowski Arango.

**Funding acquisition:** Fabio Morgante.

**Investigation:** Noah Klimkowski Arango.

**Methodology:** Fabio Morgante.

**Project administration:** Fabio Morgante.

**Resources:** Fabio Morgante.

**Software:** Noah Klimkowski Arango.

**Supervision:** Fabio Morgante.

**Validation:** Noah Klimkowski Arango, Fabio Morgante.

**Visualization:** Noah Klimkowski Arango.

**Writing – original draft:** Noah Klimkowski Arango, Fabio Morgante.

**Writing – review & editing:** Noah Klimkowski Arango, Fabio Morgante.

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
