## [Decision Letter · Decision Letter 0]

13 Aug 2024

PONE-D-24-23903Comparing statistical learning methods for complex trait prediction from gene expressionPLOS ONE

Dear Dr. Morgante,

Thank you for submitting your manuscript to PLOS ONE. After careful consideration, we feel that it has merit but does not fully meet PLOS ONE’s publication criteria as it currently stands. Therefore, we invite you to submit a revised version of the manuscript that addresses the points raised during the review process.

We look forward to receiving your revised manuscript.

Kind regards,

Ashutosh Pandey, Ph.D.

Academic Editor

PLOS ONE

Additional Editor Comments (if provided):

Reviewers' comments:

Reviewer's Responses to Questions

**Comments to the Author**

1. Is the manuscript technically sound, and do the data support the conclusions?

Reviewer #1: Yes

Reviewer #2: Yes

Reviewer #3: Yes

2. Has the statistical analysis been performed appropriately and rigorously? 

Reviewer #1: Yes

Reviewer #2: Yes

Reviewer #3: No

3. Have the authors made all data underlying the findings in their manuscript fully available?

Reviewer #1: Yes

Reviewer #2: Yes

Reviewer #3: Yes

4. Is the manuscript presented in an intelligible fashion and written in standard English?

Reviewer #1: Yes

Reviewer #2: Yes

Reviewer #3: Yes

5. Review Comments to the Author

Reviewer #1: The study compares statistical learning methods to evaluate the potential of transcriptomic data for accurate prediction of complex traits. For this purpose, the study leverages data from the Drosophila Genetic Reference Panel (DGRP) to evaluate the performance of different prediction models for predicting starvation using transcriptomic data. The study concluded that differences in prediction accuracy are minor and that methods assuming a more polygenic architecture performed better in males whereas methods performing variable selection gave higher accuracy in females. Furthermore, the results show that incorporating functional information (GO terms) can improve prediction accuracy, emphasizing the utility of integrating biological knowledge into statistical models.

Strengths

The manuscript provides an extensive comparison of ten statistical methods. By analyzing male and female data separately, the study highlights the importance of considering sex-specific genetic architectures in predictive modeling, which is crucial for traits like starvation resistance that exhibit sexual dimorphism. The results provide a thorough evaluation of differences in prediction accuracy both within sex and between sex. Furthermore, the use of Gene Ontology (GO) annotations to inform prediction models is a significant strength. Lastly, the manuscript acknowledges limitations, such as the small sample size of the DGRP lines, which limits the generalizability of the findings.

Weaknesses

The manuscript could comment on similar assessments of statistical methods for prediction of other complex traits based on transcriptomic data and whether these studies yielded similar results. Furthermore, the manuscript mentions that these methods do not account for non-linear interactions between genes, which could be crucial for predicting complex traits with epistatic effects but does not explore potential solutions to this problem. Additionally, there are some minor issues such as typos (e.g. Lines 240, 259, 291).

Reviewer #2: In this paper by Arango et al, the authors are comparing various statistical models to predict complex phenotypic traits using gene expression data. This question is fundamental to the field as many studies often employ gene expression analysis to identify gene modules using WGCNA/GWAS/TWAS. While the authors approach a very logical question, there is more information needed to strengthen this manuscript.

Major comments:

1. The authors claim that a limitation of their data is because a sample size of ~200 cell lines (~12000 highly expressed genes) is a small sample size, which is not. ~12000 genes in each sex is a decent sample size to predict phenotypic traits, especially in Drosophilia. Thus the authors must speak a bit more on why this is a limitation in the Drosophila field.

2. The authors themselves mention that the differences in prediction accuracy between the various methods is not large (see abstract and fig 1.). Thus, even if the difference exists, why would it be important to select one method over another if the difference between methods is not large? Moreover, can the authors quantify this difference to accurately determine if the difference is significant or not?

3. In lines 240-243, the authors mention that the reason the prediction accuracy was low was because of genetic variation in the lines from individual flies. However, even when such methods are applied to human data, we are dealing with high genetic variation among the population. Then in such cases why would any of these methods be employed if their accuracy is low due to genetic heterogeneity? Importantly, for Drosophilia in fact the genetic variability will be much lower (considering the lines are inbred; see line 60) than when dealing with human traits and gene expression data. This shows that such methods will ultimately will not be able to predict such variable genotypic data with accuracy. The authors should consider commenting on this.

4. In this paper the authors select starvation resistance as the phenotypic trait, however the authors state that it is well known that sex differences exists in starvation parameters (see discussion section, line 395-398). When the prediction accuracy of the statistical analysis is low, then why not also select a trait that is not known to differ by sex. This would allow to better understand if sex drives the low accuracy of the statistical parameters or if inherently these statistical tests are poor at predicting complex traits. Thus, the authors should consider comparing the methods with another trait not dependent on sex.

Minor comments:

1. The result section is written more like a discussion section. This is especially true for Gene Analysis. Please consider re-writing this section

2. Please consider proving a concluding statement for each of your result section.

Reviewer #3: The authors have presented a comparison of various statistical learning methods to predict gene expression. the manuscript is written in a clear and concise manner. The authors though can provide basis to a few parameters used (see attached manuscript). Providing relevance and basis to the parameters used for testing will enhance the depth of their work and provide more context and usefulness to the readers.

6. PLOS authors have the option to publish the peer review history of their article (what does this mean?). If published, this will include your full peer review and any attached files.

Reviewer #1: No

Reviewer #2: No

Reviewer #3: No

---

## [Author Response · Author response to Decision Letter 0]

21 Oct 2024

We have provided a detailed response to the reviewers as an attachment.

---

## [Decision Letter · Decision Letter 1]

15 Nov 2024

PONE-D-24-23903R1Comparing statistical learning methods for complex trait prediction from gene expressionPLOS ONE

Dear Dr. Morgante,

Thank you for submitting your manuscript to PLOS ONE. After careful consideration, we feel that it has merit but does not fully meet PLOS ONE’s publication criteria as it currently stands. Therefore, we invite you to submit a revised version of the manuscript that addresses the points raised during the review process.

We look forward to receiving your revised manuscript.

Kind regards,

Ashutosh Pandey, Ph.D.

Academic Editor

PLOS ONE

Journal Requirements:

Reviewers' comments:

Reviewer's Responses to Questions

**Comments to the Author**

1. If the authors have adequately addressed your comments raised in a previous round of review and you feel that this manuscript is now acceptable for publication, you may indicate that here to bypass the “Comments to the Author” section, enter your conflict of interest statement in the “Confidential to Editor” section, and submit your "Accept" recommendation.

Reviewer #2: (No Response)

2. Is the manuscript technically sound, and do the data support the conclusions?

Reviewer #2: Yes

3. Has the statistical analysis been performed appropriately and rigorously? 

Reviewer #2: Yes

4. Have the authors made all data underlying the findings in their manuscript fully available?

Reviewer #2: Yes

5. Is the manuscript presented in an intelligible fashion and written in standard English?

Reviewer #2: Yes

6. Review Comments to the Author

Reviewer #2: The authors have decently answered the concerns I raised in the previous version. However, I still have some concerns and believe that it will strengthen the paper:

1. While the authors have answered my previous comment regarding editing the result section with their justification, I still believe the part of the results section labeled as " Gene Analysis" has much background information that is not typical to Results sections. Please consider trimming this section down to keep it to the results and move parts to discussion session especially where previous studies have been compared.

2. I feel like that the authors removed the lines in 240-243 in the previous version on the basis of it not being significant to their claim and the potential for it to be confusing (mentioned in the response to the reviewer letter), but I in fact think that it is valuable when papers provide limitations of their study as no study is perfect. My suggestion would be to add this statement back and explain with the statistical equation (provided in the response letter) that the authors put in reviewer comments which will back their reasoning and avoid any confusion.

7. PLOS authors have the option to publish the peer review history of their article (what does this mean?). If published, this will include your full peer review and any attached files.

Reviewer #2: No

---

## [Editor Report · Decision Letter 2]

30 Dec 2024

Comparing statistical learning methods for complex trait prediction from gene expression

PONE-D-24-23903R2

Dear Dr. Morgante,

We’re pleased to inform you that your manuscript has been judged scientifically suitable for publication and will be formally accepted for publication once it meets all outstanding technical requirements.

Kind regards,

Ashutosh Pandey, Ph.D.

Academic Editor

PLOS ONE
---

## [Editor Report · Acceptance letter]

30 Jan 2025

PONE-D-24-23903R2 

PLOS ONE

Dear Dr. Morgante, 

I'm pleased to inform you that your manuscript has been deemed suitable for publication in PLOS ONE. Congratulations! Your manuscript is now being handed over to our production team.

Kind regards, 

on behalf of

Dr. Ashutosh Pandey 

Academic Editor

PLOS ONE